# We are What We Eat: Impact of Food from Short Supply Chain on Metabolic Syndrome

**DOI:** 10.3390/jcm8122061

**Published:** 2019-11-23

**Authors:** Gaetano Santulli, Valeria Pascale, Rosa Finelli, Valeria Visco, Rocco Giannotti, Angelo Massari, Carmine Morisco, Michele Ciccarelli, Maddalena Illario, Guido Iaccarino, Enrico Coscioni

**Affiliations:** 1Dept. of Medicine, Division of Cardiology, and Dept. of Molecular Pharmacology, Montefiore University Hospital, Fleischer Institute for Diabetes and Metabolism (FIDAM), Albert Einstein College of Medicine (AECOM), New York, NY 10461, USA; 2Dept. of Advanced Biomedical Science, Federico II University, 80131 Naples, Italy; carmine.morisco@unina.it; 3International Translational Research and Medical Education Consortium (ITME), 80131 Naples, Italy; 4Dept. of Medicine, Surgery and Dentistry, University of Salerno, 8408 Baronissi, Italy; pascalevaleria@gmail.com (V.P.); rosafinelli1@gmail.com (R.F.); valeriavisco1991@libero.it (V.V.); tintorangocico@live.it (R.G.); mciccarelli@unisa.it (M.C.); 5“San Giovanni di Dio e Ruggi d’Aragona” University Hospital, 84131 Salerno, Italy; angelo.massari@sangiovannieruggi.it (A.M.); enrico.coscioni@regione.campania.it (E.C.); 6Health’s Innovation, Campania Regional Government, 80132 Naples, Italy; illario@unina.it; 7Dept. of Public Health, Federico II University, 80131 Naples, Italy

**Keywords:** mediterranean diet, supply chain of food, metabolic syndrome, food retail, cardiovascular risk

## Abstract

Food supply in the Mediterranean area has been recently modified by big retail distribution; for instance, industrial retail has favored shipments of groceries from regions that are intensive producers of mass food, generating a long supply chain (LSC) of food that opposes short supply chains (SSCs) that promote local food markets. However, the actual functional role of food retail and distribution in the determination of the risk of developing metabolic syndrome (MetS) has not been studied hitherto. The main aim of this study was to test the effects of food chain length on the prevalence of MetS in a population accustomed to the Mediterranean diet. We conducted an observational study in Southern Italy on individuals adhering to the Mediterranean diet. We examined a total of 407 subjects (41% females) with an average age of 56 ± 14.5 years (as standard deviation) and found that being on the Mediterranean diet with a SSC significantly reduces the prevalence of MetS compared with the LSC (SSC: 19.65%, LSC: 31.46%; *p*: 0.007). Our data indicate for the first time that the length of food supply chain plays a key role in determining the risk of MetS in a population adhering to the Mediterranean diet.

## 1. Introduction

Several studies have demonstrated that the Mediterranean diet significantly reduces the risk of developing metabolic syndrome (MetS) [1,2,3,4], a cluster of clinical conditions that occur together and increase the risk of heart disease, stroke, and type 2 diabetes [5,6,7]. However, the exact role of food retail and distribution in the risk of developing MetS has not yet been fully determined.

Recently, the development of big retail food distribution has deeply modified food supply in the Mediterranean area [8,9]. Indeed, industrial retail has favored shipments of groceries from regions that are intensive producers of mass foods, generating the long supply chain (LSC) of food [10]; on the other hand, short supply chains (SSCs) involve local self-producers that promote local food markets [8]. The origin of food, the long period of time elapsing from production to consumption, the need to add preservatives, as well as the loss of perishable nutrients such as vitamins, can all contribute to reducing the quality of food. Nevertheless, whether food quality loss has an impact on the health of the population remains to be determined. 

The increasing availability of foods from big retail is a revolutionary event that has impacted health on a population-size level. In particular, the adherence to the Mediterranean diet is decreasing even within those regions where it was first discovered [11,12], and such a change in the alimentary habit is generally seen as one of the potential causes of the obesity epidemic [13], especially among adolescents [14]. 

The overarching aim of our study was to test the effects of food chain length on metabolic alterations in a population accustomed to the Mediterranean diet. Specifically, we compared SSCs of food—in which aliments are produced *in loco*, usually with traditional and low-technology methodologies—to the LSC of food. 

## 2. Methods

### 2.1. Subjects

We conducted an observational, cross-sectional study on the general population of Salerno (population: 138,000 inhabitants) and of five nearby villages (population <6000 inhabitants): Castelnuovo Cilento, Polla, Sapri, San Gregorio Magno, and Satriano di Lucania. In order to be considered eligible, subjects had to be currently and stably (for at least 10 years) living in the cities indicated, and to have signed the informed consent. 

### 2.2. Study Approval

The study was approved by the Institutional Ethical Committee of Salerno University Hospital. Written informed consent was obtained from all participants. The study is registered in the ClincalTrial.gov database (Trial number: NCT03305276).

### 2.3. Data Acquisition

On the occasion of 2015–2017 World Hypertension Day (May 17th), booths were organized in the major squares of the mentioned villages, harnessing a collaborative effort of the Medical School of Salerno and local authorities [15]. The event was successfully publicized with a 15-day notice via local media advertisements, and we had the partnership of local authorities and patient associations, as well as parishes. The population was instructed to show up on the day of the event at the booths, where subjects were asked to sign the informed consent to participate in the survey and to donate blood samples for analysis. Anamnesis and anthropometric parameters were obtained including weight, height, waist and hip circumferences, and BMI. Blood pressure was detected according to the Guidelines of the European Society of Cardiology and European Society of Hypertension (ESC/ESH) [16]. Current smokers were defined as those reporting having smoked at least 100 cigarettes during their lifetime and currently smoking every day or some days [17]. Dietary habits were collected by means of a questionnaire previously described by Trichopoulou et al. [18]. This nine-question questionnaire allows a score (Trichopoulou score) to be attributed to each subject. To determine the use of SSCs or the LSC, we formulated a questionnaire that included the following eight questions: (1) “Do the vegetables you consume come mainly from your vegetable garden?” (Yes = 1); (2) “Do you buy fruit grown in your area?” (Yes = 1); (3) “Do you eat seasonal fruit?” (Yes = 1); (4) “Does the meat you consume come mainly from local farms or from butchers in your area?” (Yes = 1); (5) “Do you eat mostly fresh, unpackaged food?” (Yes = 1); (6) “Do you eat cookies, snacks, and/or sweets more than once a week?” (No = 1); (7) “Do you use canned or frozen food?” (No = 1); and (8) “Do you drink carbonated or sweetened drinks?” (No = 1). These questions were derived from preliminary interviews performed by experienced personnel to relatives and families of volunteers in order to verify the kinds of food that are more frequently acquired from small business stores, as well as the ones mostly purchased from big food retail shops. According to this survey, fruits, vegetables, and meats were purchased more often from small business shops, whereas preserved and canned foods, as well as frozen food, sodas, cookies, and snacks, were mainly obtained from big resellers. In a second phase, we asked one big food retailer from the city of Salerno and two small business of the city of San Gregorio Magno to provide us with the suppliers of the listed products, so as to substantiate the actual length of the supply chain of food. The optimal cut-off of the score (5) was determined by receiver operating characteristic (ROC) curves (see Appendix A), applying Youden’s index [19,20]; with a score ≥5, the subject was included in the SSC group, whereas with a score <5 the subject was included in the LSC group. MetS was diagnosed according to the 2009 Harmonized Criteria, implementing the criteria of the International Diabetes Federation (IDF) to evaluate abdominal obesity [7,21].

### 2.4. Blood Sample Laboratory Analysis

A venous blood sample was collected from the antecubital vein in a dedicate booth from experienced volunteer nurses in two tubes of 5.0 mL and centrifuged the same day. The time of the last meal was recorded during data collection. We measured blood glucose, insulin, total cholesterol, HDL cholesterol, LDL cholesterol, and triglycerides. The homeostatic model assessment (HOMA) index was calculated as previously described, based on an adequate fasting time (6 h) [22].

### 2.5. Statistical Analysis

Continuous data are presented as mean ± SE. Categorical data are presented as absolute values and/or frequencies. To observe a change of one quartile in frequency with an α cut-off of 5% and a β cut-off of 20%, and given an estimated incidence of MetS of 26% in our population [23], we calculated that a *n* = 398 would have been necessary to reach statistical significance. A Kolmogorov–Smirnov test was used to verify the normality of distributions of continuous variables. A chi-square (*χ*^2^) test was used to compare frequencies. Independent sample *t*-tests were used for between-group comparisons. In all the above-mentioned tests, *p* < 0.05 was considered statistically significant. Statistical analysis was performed with SPSS (Statistical Package for the Social Sciences) 24.0 (IBM, Armonk, NY, USA) and Prism 7 (GraphPad Software, San Diego, CA, USA).

## 3. Results

### 3.1. Clinical Features of Study Population

A total of 808 subjects (45% male and 55% female, 14–85 years) were recruited during the XI, XII, and XIII editions of World Hypertension Day, which is celebrated every year on 17 May. We excluded from the analysis patients younger than 30 (*n* = 70) and older than 80 years (*n* = 30) because of the previously reported relatively low adherence to the Mediterranean diet by populations at those ages [12,24,25,26]. We also excluded those with an incomplete database, thereby precluding the calculation of adherence to the Mediterranean diet, SSCs or the LSC, or the HOMA index (*n* = 269), as well as 28 outliers (3 SD over/below mean) in MetS determinants.

The main characteristics of our population (407 subjects, 41% females, with an average age of ~56 years) are depicted in Table 1.

### 3.2. Effects of SSCs on Clinical Features

We divided the population according to the eight-point questionnaire indicated above, using the score of 5 as a cutoff to indicate adherence to SSCs (≥5) or the LSC (<5). Data are indicated in Table 1.

Our data indicate that SSCs are associated with lower levels of triglycerides and glucose, and therefore have a marked impact on the occurrence of MetS: indeed, MetS is less frequent among populations that consume SSC food. Interestingly, adherence to the Mediterranean diet, assessed using a validated questionnaire, was similar between the two populations, indicating a homogenous high adherence between the consumers of SSC and LSC foods.

### 3.3. Effects of SSCs on Insulin Sensitivity

Given the notion that MetS is a hallmark of insulin resistance, we assayed insulin resistance by means of the HOMA index. As shown in Figure 1, the HOMA index was lower in the SSC than in the LSC group (2.67 ± 0.20 vs. 4.66 ± 0.44, respectively; *p* = 0.0002).

## 4. Discussion

Our results indicate that the Mediterranean diet with food from a SSC significantly reduces the prevalence of MetS. These results are consistent with recent observations suggesting that local food environments might affect health outcomes [27]. Our data are corroborated by the evidence that insulin resistance is significantly more common among LSC subjects compared with SSC individuals. With our data, we are among the first investigators to introduce the concept that freshness of food is a key determinant of health outcomes. 

Prevalence of MetS in Southern Italy is reported to be ~25% in young adults [23] and ~65% in older women after menopause [28]; of note, the area of assessment might slightly impact the occurrence of MetS [29]. Our population showed overall a prevalence of 35%, and we consider this number to be fairly representative, given the large age range (from 30 to 80 years) of our population.

Substantial evidence indicates that local food environments might affect health outcomes [27,30,31,32,33,34,35,36]. It is therefore possible to speculate that length of a food supply chain might affect cardiovascular risk. To verify this hypothesis, it is crucial to compare populations that present similar dietary patterns that derive from different sources (i.e., retail market vs. locally grown food). In this sense, Southern Italy features examples of urbanization, where retail food is the most important source of food, which is the opposite to rural areas, where consuming locally grown, seasonal vegetables, as well as meat of courtyard animals is a fairly regular habit [28,29,37]. These contrasting local food environments, somehow superimposed on the Mediterranean diet, represent a unique setting to test the impact on health phenotypes, such as intermediate metabolism and cardiovascular risk [4,38,39].

Our findings are particularly relevant because they provide for the first time an actual evaluation of how critical the food supply chain is in the context of the Mediterranean diet. Indeed, it is well established that the Mediterranean diet can ameliorate cardiovascular risk [40]. Therefore, to verify the existence of a further improvement in health outcomes from SSCs versus the LSC, it is imperative to compare groups that consume the same diet, with no difference in macro- and micronutrients. Based on our data on insulin resistance, we can speculate that insulin sensitivity is better preserved among subjects adhering to the Mediterranean diet who eat SSC foods. Using a previously published questionnaire [18], we were indeed able to verify the high adherence of Southern Italians to the Mediterranean diet, and no differences were observed when we divided our population according to LSC and SSC groups. Therefore, we can conclude that the difference observed in terms of MetS prevalence cannot be attributable to a different attitude towards the Mediterranean diet.

Our findings should be interpreted in light of several limitations. First and foremost, the cross-sectional design of the study prevents a determination of causality. Another major issue that was not addressed by our questionnaires is the lifestyle that accompanies the provision of SSC food, nor did our questionnaire did not measure the time spent in the production of such food, such as the time spent looking after crops and/or courtyard animals, which would imply a more active lifestyle. Likewise, the use of food from the LSC is more distinctive of urban centers, where alternative healthy lifestyles are also more common (practicing sports, attending the gym). This aspect was also testified by our observation of no significant differences in body weight or waist between the two populations, with a tendency of heavier weights detected in the SSC group. Therefore, while we cannot completely rule out an effect of physical activity on MetS in our study, the evidence that SSC is associated with a lower prevalence of MetS is suggestive of a fundamental impact of fresh food on metabolic parameters.

If confirmed in larger prospective studies, these results could promote public interventions to improve lifestyle, preferring, whenever possible, SSCs to the LSC. Therefore, an assessment of cardiovascular risk should include dietary habits of the studied population.

## 5. Conclusions

Taken together, our findings indicate for the first time that the length of food supply chain is crucial in determining the risk of developing MetS as well as in the assessment of cardiovascular risk in a population adhering to the Mediterranean diet.

## Figures and Tables

**Figure 1 jcm-08-02061-f001:**
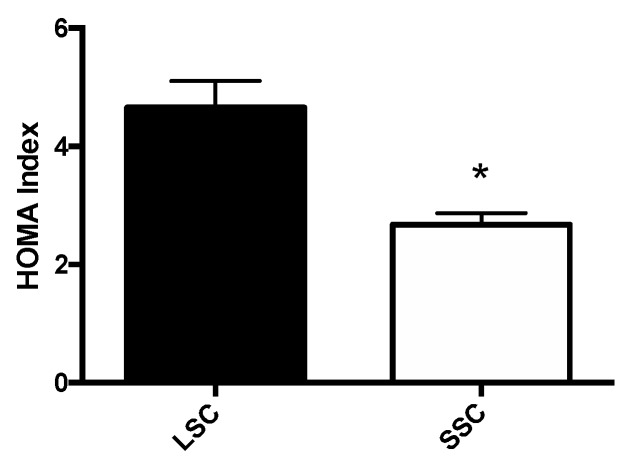
Impact of SSCs and the LSC on insulin resistance. LSC: Long supply chain; SSC: Short supply chain; * *p* = 0.0002.

**Table 1 jcm-08-02061-t001:** Impact of SSCs and the LSC on anthropometric and clinical characteristics.

	Total	LSC	SSC	*p*
***N***	407	178	229	-
**Age (years)**	55.9 ± 0.58	56.4 ± 0.8	55.52 ± 0.8	0.422
**Sex (M, %)**	59	60	58	0.765
**Weight (Kg)**	73.8 ± 0.82	72.2 ± 1.2	75.1 ± 1.08	0.085
**Height (cm)**	163.7 ± 0.6	164.2 ± 0.6	163.3 ± 0.91	0.399
**Waist (cm)**	96.3 ± 0.74	96.4 ± 0.84	96,0 ± 1.49	0.803
**BMI (Kg/m^2^)**	27.6 ± 0.25	27.1 ± 0.39	27.9 ± 0.34	0.098
**SBP (mmHg)**	130.6 ± 0.9	131.2 ± 1.3	130.1 ± 1.2	0.523
**DBP (mmHg)**	79.8 ± 0.5	80.5 ± 0.8	79.2 ± 0.69	0.220
**HR (bpm)**	72.2 ± 0.6	72.1 ± 0.8	72.28 ± 0.82	0.877
**Fasting Glucose (mg/dl)**	84.4 ± 1.2	91.28 ± 1.7	79.41 ± 1.5	0.001
**Serum Insulin (μU/dl)**	17.7 ± 0.97	21.4 ± 1.7	14.9 ± 1.1	0.001
**Creatinine (mg/dl)**	0.85 ± 0.02	0.88 ± 0.05	0.82 ± 0.01	0.19
**Current Smokers (%)**	32.0	30.0	34.0	0.427
**Cholesterol (Total, mg/dl)**	201.4 ± 1.9	201.36 ± 3.2	201.48 ± 2.5	0.977
**Cholesterol (HDL, mg/dl)**	59.3 ± 0.7	59.03 ± 1.2	59.56 ± 1.0	0.737
**Cholesterol (LDL, mg/dl)**	124.6 ± 2.1	127.02 ± 3.9	123.43 ± 2.4	0.418
**TG (mg/dl)**	121.7 ± 3.6	136.14 ± 5.9	110.95 ± 4.3	0.001
**Metabolic Syndrome (%)**	24.81	31.46	19.65	0.007
**Trichopoulous Score**	4.98 ± 0.08	4.86 ± 0.13	5.08 ± 0.10	0.180

Frequencies are reported as %, continuous variables as mean ± SE; DBP: Diastolic/systolic blood pressure; HR: Heart rate; HDL/LDL: High-density/low-density lipoproteins; LSC/SSC: Long/short supply chain; TG: Triglycerides; Trichopoulous Score: Score of the adherence to a Mediterranean-style diet (9 = max, 0 = min; *p* value was calculated applying the *t* test or χ^2^, as appropriate).

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
