# Peer review of "We are What We Eat: Impact of Food from Short Supply Chain on Metabolic Syndrome"

_jcm, 2019, doi:10.3390/jcm8122061_

Round 1

Reviewer 1 Report

I thank the authors for the substantial changes and improvement of the manuscript. However, I still have some doubts and concerns which I am stating below:

Abstract

Average or mean age cannot be just a number. Please, present age as median and interquartile range. If using average/mean, the standard deviation should be stated as well.

Methods

I am satisfied with author's more detailed description of data acquisition. However, I still can't see the eligibility criteria, the sources and methods of selection of participants and calculation of the size of study sample. Also, it is very important to justify the validity of the questionnaire. Please, give more details on the development and validation process of the questionnaire (any published work?). Why was score of 5 chosen as a cut of?

Data reported in results should be explained in methods, ie. please explain Trichopoulous score, define smokers..

Results

BMI- provide unit.

Table 2. t-test (as stated in the table) is not appropriate test for all the analysis.

Report exact p value for the difference of SSC and LSC on insulin sensitivity. Also the value (not seen the exact value from the graph)

The sample is very heterogenic (ie. large age range)- could you explore if your findings apply to all the age groups?

Discussion

Please discuss limitations of the study.

Author Response

I thank the authors for the substantial changes and improvement of the manuscript. However, I still have some doubts and concerns which I am stating below:

R: We thank this Reviewer for her/his words of appreciation towards our work and for the valuable feedback.

Abstract. Average or mean age cannot be just a number. Please, present age as median and interquartile range. If using average/mean, the standard deviation should be stated as well.

R: We are now presenting age as mean±SD, as requested.

Methods. I am satisfied with author's more detailed description of data acquisition. However, I still can't see the eligibility criteria, the sources and methods of selection of participants and calculation of the size of study sample. Also, it is very important to justify the validity of the questionnaire. Please, give more details on the development and validation process of the questionnaire (any published work?). Why was score of 5 chosen as a cut of? Data reported in results should be explained in methods, ie. please explain Trichopoulous score, define smokers.

R: We thank this Reviewer for her/his insightful suggestions. In the revised version of the manuscript, we expanded the methods section, providing all the requested information: eligibility, power analysis performed to calculate the sample size, a new supplementary figure showing the analysis of the ROC curves and the application of the Youden’s test used to determine the score cutoff (previously only mentioned in the methods section), an explanation of the Trichopoulous score, a definition of current smokers (defined as those reporting having smoked at least 100 cigarettes during their lifetime and currently smoking every day or some days).

Results. BMI- provide unit.

R: Fixed, thanks.

Table 2. t-test (as stated in the table) is not appropriate test for all the analysis.

R: Fixed, thanks.

Report exact p value for the difference of SSC and LSC on insulin sensitivity. Also the value (not seen the exact value from the graph)

R: We now report the exact p value, as requested.

The sample is very heterogenic (ie. large age range)- could you explore if your findings apply to all the age groups?

R: We thank this Reviewer for this interesting comment. As indicated in the methods, the numerosity of our population was tailored on the hypothesis that the impact of SSC could cause a -20% reduction of the rate of MetS. For this reason, the application of the analysis to subgroups of our population might result in rejected hypothesis just for the lack of enough power. Indeed, this might be the case. We divided our population in two halves, using 55 years as age cut-off. Indeed, the chi square analysis confirmed the hypothesis in the younger group (<55 years, prevalence of MetS: LSC vs SSC, 31,48% vs 17,57, p<0.05), but not in the older group (>55 years, prevalence of MetS: LSC vs SSC, 36,07% vs 25,00, p<0.11). Given the low value of the p, we believe that this lack of statistical significance is mainly due to a not large enough number of cases. For this reason, we prefer to leave this analysis out of the paper, in order to avoid confusing interpretations.

Discussion. Please discuss limitations of the study.

R: We have added a paragraph to discuss the limitations of our study.

Reviewer 2 Report

In the present manuscript, authors have investigated the impact of type of food retail and distribution on the prevalence of metabolic syndrome in a Mediterranean diet-adhering population. This is new, re-submitted version of the manuscript, as former version was written poorly. Present manuscript offers major improvements in total writing impression, as well as methods, results and discussion sections. However, I still believe that, in order for this paper to be publishable, some major improvements are still needed, and main remark must be cleared.

More clarification is needed for the criteria of the “adherence to Mediterranean diet”. Detailed explanation is needed in “methods” section. Also in Methods section part about laboratory analysis is not adequate. Was blood sampling performed in fasting conditions? Which procedures were used? Information about HOMA calculation is missing (with the appropriate reference). More clarification is needed about the criteria used for MetSy diagnosis (I also think that reference stated in the manuscript is wrong, it is stated that criteria is defined with ref 7?) Furthermore, you do not present data about waist circumference? Why have you excluded patients younger than 30, and older than 80? Lines 152-154: in the cross-sectional design of the study, it is not acceptable to have causal conclusions like this (only speculating in discussion). English language needs improvement in the whole manuscript. Also, it is better to refer to HOMA index as an insulin resistance measure, not insulin sensitivity. For HOMA index it is needed to write exact values for both groups in the text (presented in Fig 1). Table 1 and 2 could be combined in one table (all, LSC, SSC, p value)  I strongly believe that lack of information about the lifestyle of subjects included (esp. physical activity) is a very important limitation and it is not enough to just mention in Discussion section this as one of the limitations. Because of this limitation, I am not confident with the results and conclusions of this study. Discussion section should be improved, because in depth discussion about the results and wider context is still limited. Additionally, the limitations section needs extension.

Round 2

Reviewer 1 Report

I thank the authors for accepting my suggestions and for answering all my doubts. In my opinion authors did substantial changes of the manuscript what makes article acceptable for publishing. Although, I am not qualified to judge on English language, I noticed some language problems and would recommend additional editing.

Author Response

We thank this Reviewer for her/his insightful comments and suggestions, which considerably ameliorated the quality our manuscript. We agree with your last observations and we apologize for the typos and errors contained in the previous version of the manuscript. We have corrected all the typos and improved grammar and syntax; furthermore, as requested, the paper has been proofread by a native English speaker.

This manuscript is a resubmission of an earlier submission. The following is a list of the peer review reports and author responses from that submission.

Round 1

Reviewer 1 Report

This study has so many flaws, from design to interpretation and is below level of acceptance for the publication. I will mention some of them: rationale of the study is limited, data about the selection of the study population, data about physical activity is missing, authors are mentioning Mediterranean diet so many times (did they tested adherence to MD in the studied population), the questionnaire used for testing is not adequate, information about blood drawings and test used is missing, in Results section almost whole paragraph is even not a part of the study results, discussion very short without any real discussion, conclusions are beyond the scope of the manuscript,…  

Reviewer 2 Report

This is an interesting article which provides an insight into the effects of food chain length on the prevalence of MetS in a population accustomed to Mediterranean diet. The research question is original and to the best of my knowledge was not investigated earlier. However, methodology and results are deficient. I felt that both were a bit rushed and some attention in terms of clarification and providing more details should be used.  Please see specific comments below:

Introduction

Pg 1, ln 30 – please check the references 1-7 if they really support the evidence that Mediterranean diet significantly reduces the risk of developing MetS. Up to date there are lot of studies with higher level evidence (systematic reviews and randomisation studies) on this topic. Also, please include only the studies whose aim was to investigate the influence of Mediterranean diet on MetS.

Methods

Pg 2 ln 45-46: please provide more details on the eligibility criteria, and the sources and methods of selection of participants. Explain how the study size was arrived at.

Pg 2 ln 52: could you describe: “Boots were organized to collect anamnesis and dietary habits” How and where were boots organized? Which data did you collect? How did you collect data on dietary habits? Did you use any food frequency questionnaire?

Pg 2 ln 53: more details needed on the development of the questionnaire. Validation of the questionnaire?

Results

Data on dietary habits should also be presented in the results. It is very important to show that both groups (short and long chain food supply groups) consume same diet with no difference in macro and micronutrients. Otherwise, the influence on MetS prevalence could not be contributed solely to the difference in food chain length. If authors collected this data properly it should be analysed and shown in the results. Otherwise, I consider it as an important limitation of the study.    

In general, results section does not provide much data and authors should rethink what would be important to present to get better understanding of the study.

Pg2 ln 76-80: not part of the results section. Would be better in discussion

Discussion

Would be good to discuss …What is the prevalence of MetSy in general population in southern part of Italy? Is the sample representative? What are the differences in the lifestyle for the two groups? Do people who consume food from short chain supply have different lifestyle (i.e. more physical activity, less sedentary lifestyle) which could also influence prevalence of MetS? Any other sociodemographic differences which could influence? More variables should be reconsidered and discussed.